# A Systematic Review and Meta-Analysis Comparing the Effectiveness of Transversus Abdominis Plane Block and Caudal Block for Relief of Postoperative Pain in Children Who Underwent Lower Abdominal Surgeries

**DOI:** 10.3390/medicina59091527

**Published:** 2023-08-24

**Authors:** Dan Xiao, Yiyuan Sun, Fang Gong, Yu Yin, Yue Wang

**Affiliations:** 1Department of Pediatrics, Yongchuan Hospital Affiliated to Chongqing Medical University, Chongqing 402160, China; sunxxxd@126.com (D.X.); eileen0606@163.com (Y.W.); 2Day Surgery Center, General Practice Medical Center, West China Hospital, Sichuan University, Chengdu 610017, China; drsyyrmb@126.com (Y.S.); 15228845721@163.com (Y.Y.)

**Keywords:** pain scores, abdominal surgery, caudal block (CB), transversus abdominis plane block (TAPB), meta-analysis

## Abstract

*Background and Objectives*: Postoperative pain after lower abdominal surgery is typically severe. Traditionally, in pediatric anesthesia, a caudal block (CB) has been used for pain management in these cases. Nowadays, a transversus abdominis plane block (TAPB) seems to be an effective alternative. However, which technique for perioperative analgesia is better and more effective remains unclear in children who undergo abdominal surgeries. The aim of this study was to compare the efficacy and safety of a TAPB and CB for pain management in children after abdominal surgery by conducting a meta-analysis of published papers in this area. *Methods*: We conducted a thorough search of PubMed, EMBASE, the Cochrane Library, and the Web of Science for randomized controlled trials (RCTs) that compared a TAPB and CB for pain management in children who had abdominal surgery. Two researchers screened and assessed all the information with RevMan5.3 used for this meta-analysis. Pain scores, the total dose of rescue analgesic given, the mean duration of analgesia, the intraoperative and postoperative hemodynamic conditions 24 h after surgery, and adverse events were compared. *Results*: 15 RCTs that involved a total of 970 pediatric patients were included in this study. The results of this meta-analysis showed that there were no significant differences between the 2 groups in terms of postoperative pain scores at 1 h (SMD = 0.35; 95% CI = −0.54 to 1.24; *p* = 0.44, I^2^ = 94%), 6 h (SMD = −0.10; 95% CI = −0.44 to −0.23; *p* = 0.55, I^2^ = 69%), 12 h (SMD = −0.02; 95% CI = −0.45 to −0.40; *p* = 0.93, I^2^ = 80%), and 24 h (SMD = −0.66; 95% CI = −1.57 to −0.25; *p* = 0.15, I^2^ = 94%); additional analgesic requirement (OR 0.25; 95% CI 0.09 to 0.63; *p* = 0.004, I^2^ = 72%); total dose of rescue analgesic given in 24 h (SMD = −0.37; 95% CI = −1.33 to −0.58; *p* = 0.44; I^2^ = 97%); mean duration of analgesia (SMD = 1.29; 95% CI = 0.01 to 2.57; *p* = 0.05, I^2^ = 98%); parents’ satisfaction (SMD = 0.44; 95% CI = −0.12 to 1.0; *p* = 0.12; I^2^ = 80%); and intraoperative and postoperative hemodynamic conditions 24 h after the surgery and adverse events (SMD = 0.78; 95% CI = 0.22 to 2.82; *p* = 0.70; I^2^ = 62%). Compared to a CB, a TAPB resulted in a small but significant reduction in additional analgesic requirement after surgery (OR 0.25; 95% CI 0.09 to 0.63; *p* = 0.004). *Conclusions*: TAPBs and CBs result in similar efficient early analgesia and safety profiles in children undergoing abdominal surgeries. Moreover, no disparities were observed for adverse effects between TAPBs and CBs.

## 1. Introduction

Various types of abdominal surgeries are performed on children, which may include but are not limited to surgery for inguinal hernias, laparoscopic surgery of the abdomen, and surgical procedures for the urinary system. Patients who undergo abdominal surgery often suffer severe postoperative pain, which can prevent recovery and trigger a variety of pathophysiological changes [1]. An effective and safe pain management model is therefore very important for patients during the perioperative period. Effective pain relief can be provided by classical postoperative analgesia following surgery, although this is associated with a well-defined risk of adverse effects [2]. Regional anesthesia has therefore become a key part of multimodal analgesic therapy due to it enhancing recovery after surgery.

Regional anesthesia comprises both central and peripheral techniques. One of the most common techniques of regional anesthesia used in children is a caudal epidural block (CB) [3], while peripheral blocks, such as a transversus abdominis plane, rectus sheath, and ilioinguinal/iliohypogastric, are becoming popular means of providing analgesia for abdominal procedures [4,5]. For painful subumbilical interventions, CBs are an effective way of providing perioperative analgesia and account for 34–40% of patients receiving pediatric regional anesthesia [6]. However, caudal blocks have been linked to hemodynamic/systemic or local adverse events, with some of the potential risks associated with the procedure including arrhythmia, hypotension, respiratory depression, seizures due to toxicity, inflammation or infection of the puncture site, sacral osteomyelitis, or injury to the local nerves [7]. Currently, visual access and confirmation by ultrasound (US) have resulted in a decrease in these side effects associated with CBs [8]. Another analgesic technique in abdominal surgery described by Rafi [9] was the transversus abdominis plane block (TAPB). This technique has now become one of the most frequently performed regional anesthesia procedures [10] and has provided relief from postoperative pain in children undergoing open and laparoscopic abdominal surgery. However, the application of a TAPB is still very complex despite appearing simple in theory. This is because targeting the transversus abdominis plane compartment requires different techniques and methods, with the size of the area requiring careful calculation of the local anesthetic dose in order to properly manage postoperative pain. In addition, the majority of approaches for a TAPB only treat somatic pain in the abdominal wall and not pain related to the internal organs [10].

With the exception of the study by Bryskin et al. [11] in which a fixed postoperative analgesia regimen was used, all other studies describe waiting for breakthrough pain despite it not being discussed in any previous paper. Due to clinical and ethical standards, it has been difficult and not possible to conduct large-scale, multicenter, randomized controlled trials (RCTs) in the child population to compare which of the two technologies is better and more effective. Therefore, the aim of the current review was to gather available evidence for the use of CBs and TAPBs in pediatric anesthesia and to provide guidance for further investigations. This review also evaluated the efficacy of CBs and TAP blocks reported by RCTs in reducing pain and improving recovery following abdominal surgeries in children and adolescents.

## 2. Methods

### 2.1. Electronic Literature Search Strategy

In order to identify RCTs that compared the use of CBs and TAPBs for analgesia after abdominal surgery in children, a literature search of electronic databases including the Web of Science (1945-present), the Cochrane Library (2000-present), Embase (1947-present), and PubMed (1966-present) was conducted by a librarian on 28 February 2023. The search terms used were “transversus abdominis plane block”, “TAP”, “caudal block”, “CB”, “abdominal surgery”, and “children”. The complete search strategies for each database can be found in Appendix A.

### 2.2. Data Extraction

Two reviewers independently screened the titles, abstracts, and keywords using inclusion criteria. Any differences were settled by discussing until a common ground was found. One of the reviewers then looked through the references manually.

### 2.3. Criteria for Considering Studies for the Review

The PROSPERO protocol registration number CRD42023389280 was assigned for the review protocol, which can be found in the International Prospective Register of Systematic Reviews. The submitted report adhered to the Preferred Reporting Items for Systematic Reviews and Meta-analyses (PRISMA) guidelines.

### 2.4. Types of Studies

Pediatric participants aged between 0 and 18 years were included in this analysis as long as the study in which they were included was an RCT with no limitations on language or year. This study included any level of CB vs. TAPB for surgical procedures mainly involving infraumbilical abdominal incisions, such as orchiopexy, inguinal, appendectomy, Meckel diverticulum, testicular detorsion, and hydrocelectomy.

The pain scores were the main focus of this study, while additional parameters included the time when rescue analgesia was needed, the total amount of rescue analgesics administered, the average duration of pain relief, satisfaction levels, and any negative impacts, like postoperative nausea or other complications.

### 2.5. Data Extraction and Management

The data from each study included in this review were extracted by two investigators and comprised fundamental information (author’s name, published year, location, sample size, age of the patients, type of surgery, anesthetic technology, background analgesia rescue analgesia, and follow-up), primary outcomes (postoperative pain score at 1, 6, 12, and 24 h), and secondary outcomes (intraoperative and postoperative hemodynamic conditions, additional analgesic requirement, total dose of rescue analgesic given in 24 h, time to first given analgesia, duration of postoperative analgesia, parents’ satisfaction, and adverse effects). The pain scores were recorded using either visual or numeric scales. The measures of time in the data were uniformly converted into hours. Median and extreme data were estimated and expressed as the mean and standard deviation. For data presented in a different format than mean and standard deviation, we reached out to the author to obtain the original data. In cases where this was not feasible, the data were transformed into average and typical deviations using the methodology described earlier. This enabled incorporation of additional studies into this meta-analysis [12].

### 2.6. Assessment of the Risk of Bias in the Included Studies

Two reviewers independently assessed each study for bias using the *Cochrane Reviewer’s Handbook* guidelines. The possible sources of bias, such as random sequence generation, allocation concealment, blinding of participants, personnel and outcome assessors, incomplete outcome data, and selective reporting, were examined in all the studies. Differences in assessment were agreed upon through discussions. Attrition bias was considered if the loss to follow-up rate exceeded 10%. RevMan for Windows version 5.3, a tool provided by the Cochrane Collaboration in Oxford, UK, was used to conduct this meta-analysis. The summary measures were the standardized mean difference for pain score and the mean difference in time until the use of supplementary analgesia.

Table 1 presents the potential bias risk (methodological heterogeneity) and variations in study design. The I^2^ statistic was used to calculate statistical heterogeneity, which represented the variability in the estimate of each effect that could be attributed to true differences in the studies. We classified <30% as low, 30–60% as moderate, and >60% as high variability. A fixed effects model was chosen unless I^2^ was ≥50% where a random effects model was chosen instead. *p* values < 0.05 were considered statistically significant. If the *p*-value was <0.05 and the 95% CI for OR was not equal to 1, then the SMD or OR was considered statistically significant. The *Cochrane Reviewer’s Handbook* method was used to transform raw data reported as 95% CIs into SDs via the inverse of the t distribution for synthesis.

### 2.7. Assessment of Heterogeneity

Two investigators evaluated the methodological quality of each RCT using the *Cochrane Handbook*. In cases where there was disagreement, the final decision was made by a third researcher. During the assessment, factors such as generation of random sequence, concealment of allocation scheme, blinding, precision of data results, absence of selective reporting, and other prejudices were evaluated.

## 3. Results

A total of 15 RCTs comprising a total of 970 participants were included in this study [11,13,14,15,16,17,18,19,20,21,22,23,24,25,26]. All the papers were published in English. The age range of all the participants was under 18 years. Table 1 contains a comprehensive breakdown of the attributes of the studies, while Figure 1 illustrates both the procedure and the findings of the literature examination. A summary of the impact of the results of this meta-analysis is contained in Table 2.

### 3.1. Risk of Bias

*The Cochrane Handbook* for systematic reviews of interventions was used to assess the risk of bias in the RCTs (Figure 2). Jadad scores were applied to estimate this risk (Table 1). A modified Jadad scale score of 4 or greater was considered to indicate high quality, while a score of 0 to 3 was considered indicative of low quality. The assessment of risk of bias in each domain for all the studies is summarized in Figure 3. Ten studies [11,13,15,16,18,19,22,23,25,26] used random number tables or computer-generated random numbers, one study [14] used sealed envelopes, while the random sequence generation was not described in four studies [17,20,21,24]. Five studies [15,16,18,19,22] recorded that treatment allocation was sealed in an opaque envelope, while one study used a random allocation table [11]. The method used to blind the subjects was not mentioned in nine studies [13,14,17,18,20,22,23,24,25]. Nine studies [11,13,14,15,16,17,21,24,25] used blinding in order to measure outcomes, while six studies did not. However, there was little potential for a substantial impact on the results of all the patients who were children who underwent a general anesthesia operation with no reporting bias being detected. Furthermore, all trials indicated that they were completed without any participant withdrawals. Significant levels of other biases were not identified in any of the 15 trials.

All RCTs in this study detailed the performance of the CB or TAPB. However, due to variability in the criteria for determining adequate analgesia, the final outcome could be subject to bias. We identified differences in the type and amount of local anesthetics administered, although it remains uncertain whether this variation impacted the end result. Pain was measured using varying scales in the studies, with subjective judgment being particularly prevalent among children. This may have resulted in measurement bias. Moreover, differences emerged in terms of disease and the surgeries undertaken in the trials, and it is possible that these differences may have introduced potential biases in our meta-analysis. We were also unable to conduct a subgroup analysis due to insufficient data. To ensure thoroughness in our research, two researchers (HT and YY) independently screened related publications. Furthermore, we broadened our search criteria by including additional items and keywords in our search of databases, conference records, and registered trials. Although we recognize that publication bias might have impacted our meta-analysis because studies with unfavorable outcomes are less likely to be published, this may have led to an overestimation of this impact.

### 3.2. Pain Scores

All studies reported pain scores. Postoperative pain scores after abdominal surgery were assessed with the CHEOPS (Children’s Hospital of Eastern Ontario Pain Scale) score in five studies [13,17,18,24,25], the FLACC (Face, Legs, Activity, Cry, and Consolability) score in eight studies [11,14,15,16,19,20,23,26], and the objective pain scale in two studies [21,22].

### 3.3. Postoperative Pain Scores at 1 h

A total of 3 studies [13,18,25] including 212 patients reported pain scores 1 h after abdominal surgery using the CHEOPS score. Significant heterogeneity observed between the studies (I^2^ = 99%, *p* < 0.001) led to the utilization of a random effects model. The groups did not show any difference in their postoperative pain scores 1 h post-surgery (SMD = 1.83; 95% CI = −1.46 to 5.12; *p* = 0.28; Figure 4). Seven studies [11,15,16,19,20,23,26] including 459 patients reported pain scores 1 h postoperatively using the FLACC score. As shown in Figure 5, there was significant heterogeneity (I^2^ = 94%) but no difference between the scores (SMD = 0.35; 95% CI = −0.54 to 1.24; *p* = 0.44).

### 3.4. Postoperative Pain Scores at 6 h

FLACC pain scores were reported 6 h after abdominal surgery in 8 studies [11,14,15,16,19,20,23,26] involving 519 patients. In order to account for the significant heterogeneity observed among the studies, a random effects model was implemented (I^2^ = 69%, *p* < 0.001). Six hours post-surgery, the two groups had identical scores for postoperative pain (SMD = −0.10; 95% CI = −0.44 to −0.23; *p* = 0.55; Figure 5).

### 3.5. Postoperative Pain Scores at 12 h

A total of 459 patients in 7 studies [11,15,16,19,20,23,26] underwent abdominal surgery and reported FLACC pain scores after 12 h. Due to significant heterogeneity among the studies, a random effects model was applied (I^2^ = 80%, *p* < 0.001). The comparison between the two groups showed no significant difference in postoperative pain scores after 12 h (SMD = −0.02; 95% CI = −0.45 to −0.40; *p* = 0.93; Figure 5).

### 3.6. Postoperative Pain Scores at 24 h

A total of 7 studies [11,14,15,16,20,23,26] including 459 patients who underwent abdominal surgery reported FLACC pain scores after 24 h. The application of a random effects model was deemed necessary due to substantial heterogeneity detected among the studies (I^2^ = 94%, *p* < 0.001). At 24 h after the operation, both groups had the same postoperative pain scores with no significant difference observed (SMD = −0.66; 95% CI = −1.57 to −0.25; *p* = 0.15; Figure 5).

The POAS scores in one study reported no difference between the two groups [21]. However, the values of MOPS in another study were significantly lower in the TAPB group than in the CB group at 8 h postoperatively (*p* < 0.05) [22]. Finally, in one study, none of the children assessed with the Bieri Faces pain scale [27] experienced pain two months after surgery.

### 3.7. Additional Analgesic Requirement

A total of 7 studies [14,15,16,17,18,19,21] including 439 patients reported additional analgesic requirements 0–24 h after abdominal surgery. TAPBs showed a considerable advantage in terms of the standardized mean difference (OR 0.25; 95% CI 0.09 to 0.63; *p* = 0.004, Figure 6), although the inconsistency was high (I^2^ = 72%).

### 3.8. Total Dose of Rescue Analgesic Given in 24 h

A total of 9 studies [13,15,16,19,21,22,24,25,26] including 673 patients reported this outcome. As a result of detecting substantial heterogeneity among the studies, a random effects model was utilized (I^2^ = 97%, *p* < 0.05). Overall, no difference was observed between the two groups for this outcome (SMD = −0.37; 95% CI = −1.33 to −0.58; *p* = 0.44; Figure 7).

### 3.9. The Mean Duration of Analgesia

The length of time for pain relief after surgery was determined by measuring the period from extubation until the first instance of needing pain medication. A total of 9 studies [13,14,15,16,20,21,22,24,25] including 671 patients reported the average length of time that pain relief lasted after abdominal surgery. As significant heterogeneity was observed among the studies, a random effects model was utilized (I^2^ = 98%, *p* < 0.10). The two groups showed no noticeable dissimilarity in the mean of the duration of analgesia (SMD = 1.29; 95% CI = 0.01 to 2.57; *p* = 0.05; Figure 8).

### 3.10. Intraoperative and Postoperative Hemodynamic Conditions

A total of 11 studies [13,15,16,17,18,19,20,21,22,24,25] including 763 patients reported this outcome with no study recording any difference between the 2 groups. A total of 1 study reported that the mean heart rate 6 h postoperatively was significantly higher (*p* = 0.045) in the CB group (87.84 ± 11.73 bpm) than that measured in the TAPB group (82.48 ± 8.52 bpm) [23]. The other three articles did not mention vital parameters [11,14,18].

### 3.11. Parents’ Satisfaction

A total of 4 studies [14,16,18,23] including 282 patients reported this outcome. Significant heterogeneity among the studies led to the application of a random effects model (I^2^ = 80%, *p* < 0.05). Overall, no difference was observed between the two groups for this outcome (SMD = 0.44; 95% CI = −0.12 to 1.0; *p* = 0.12; Figure 9). One study with incomplete data also indicated that there was no statistically significant difference between the groups for the satisfaction levels reported by parents and surgeons [21].

### 3.12. Adverse Effects

A total of 5 studies [16,19,23,24,25] including 352 patients reported adverse effects after abdominal surgery. The studies exhibited moderate heterogeneity, leading to the use of a random effects model (I^2^ = 62, *p* = 0.03). The occurrence of adverse events was similar in both groups (SMD = 0.78; 95% CI = 0.22 to 2.82; *p* = 0.70; Figure 10). The remaining ten studies did not report adverse reactions in the two groups.

## 4. Discussion

Overall, the results of our review suggest that when compared with a CB, a TAPB appeared to have no detectable differences in pain release in patients within 24 h after surgery. No patients experienced pain in the second postoperative month in Polat’s report [14]. Similarly, no significant differences were observed in either the total dose of rescue analgesic given, the mean duration of analgesia, or the intraoperative and postoperative hemodynamic conditions 24 h after surgery. However, the study data showed that the TAPB group had lower additional analgesia requirements 24 h after surgery. No noticeable differences in adverse effects or the contentment of parents were observed between CBs and TAPBs.

Our meta-analysis showed that a TAPB, as a relatively new technique, was not clinically different from a CB, which is the regional analgesia used traditionally in pediatric surgery [28,29]. A meta-analysis found that a CB provides better analgesia and requires less rescue analgesic dose in children compared to non-caudal regional anesthesia [30]. However, the incidence rates of urinary retention and motor blocks are higher in the CB group [30]. Moreover, another meta-analysis suggested that TAPBs can provide a longer analgesia duration than CBs for pediatric inguinal and genitourinary surgeries [31]. Bryskin and colleagues reported that a TAPB was less effective than a CB for reducing the severity of bladder spasms when used in 45 children undergoing ureteral reimplantation surgery [11]. The studies included in our meta-analysis indicated that both techniques alleviated pain associated with lower abdominal surgery. Each doctor’s considerations for patients vary. From a clinical viewpoint, the successful prospect of a CB performed on mid-abdominal surgeries, such as an umbilical hernia repair, in children is unclear [3]. In addition, contraindications in children that may affect the blocking effect include a history of abdominal surgery, local site infection, trauma, or the presence of congenital diseases [3]. Hence, this finding could provide a high level of evidence for these considerations of regional anesthetic techniques in pediatric lower abdominal surgeries.

There is a strong association between adolescent opioid and NSAID misuse and subsequent drug-related disorders and also high-risk behavior that persists into young adulthood [32]. Reduction in rescue analgesics is therefore generally desirable and necessary in children and adolescents, particularly in vulnerable populations. Techniques of regional anesthesia after surgery greatly reduced the dose of rescue painkillers. An inability to place the block or block failure is the most frequently observed adverse event [29]. Using US and unfamiliar applications may also lead to a longer processing time for TAPBs or CBs due to image adjustments. A TAPB in bilateral surgeries requires two separate blocks, which also leads to a longer anesthesia period. Moreover, major complications of US-TAPB are rare. However, there are reports in adult literature that have described some related complications, such as transient femoral nerve palsy, peritoneal puncture, organ injury, and postoperative urinary retention [33,34,35,36]. The incidence of these complications is unknown. In contrast, a CB is easier to perform and be mastered by an anesthetist. A success rate for a blind technique above 96% was achieved in children [37]. However, the limited success rate in CBs for mid-abdominal surgical interventions has been shown to be due to the unpredictable secondary spread of local anesthetics and a significant inverse relationship between age, weight, and height [38,39]. Although a CB is related to hemodynamic and systemic or local adverse events, it has a very low incidence rate [3]. Polaner et al. reported the caudal group had no complications for a limited duration [3], while another study demonstrated that central nerve blocks and the use of a catheter were risk factors for regional anesthesia in children younger than 6 months [3,40].

After critical analysis of data in our review, it could clearly be concluded that both TAPBs and CBs were efficient and enabled periprocedural hemodynamic stability. A large multicenter European study also demonstrated that regional anesthesia in pediatric patients was extremely safe, with an overall low complication rate of 0.12% [40]. Our analyses confirmed that the 2 techniques were effective and safe to use in combination with ultrasound, with only a few (5.1%) mild complications observed in all 970 patients in the review. Furthermore, there were no significant postoperative complications, such as nausea, vomiting, or urine retention. The administration of dexamethasone as an addition to the local anesthetic has increased, as it aids in extending the duration of pain relief and ultimately lessens the amount of analgesics needed for rescue purposes [20]. However, these findings mainly focused on adult patients and did not involve the pediatric population.

Our review had several limitations due to the low quality of the original data. First, the rescue analgesic drugs used were different between studies. Opioids or NSAIDs were selected as rescue analgesics according to the needs of patients but caused undesirable effects, such as vomiting, itchiness, and, while rare, life-threatening respiratory depression. These outcomes may have contributed to the adverse effects associated with the TAPB or CB. In addition, the surgeons and anesthesiologists did not use standardized methods for tissue handling and local analgesia. Subgroup analysis data for the different surgeries could also not be obtained, while open or minimally invasive surgery may have affected overall outcomes. Moreover, some studies did not describe the exact methods of randomization, allocation, and even blinding for participants or outcome assessments, but all the included trials assessed moderate or high Jadad scores. The disparity in pain scores and secondary results may be attributed to the absence of consensus in pain evaluation measures across multiple investigations. In addition, there was variability in the use and volume of local anesthetics that varied between trials. Different evaluation tables for outcome indicators were used in these studies, resulting in high heterogeneity. Due to the considerable diversity in research techniques, we could not evaluate the overall quality of the evidence. Ropivacaine has been proven to be more effective than bupivacaine for ilioinguinal blocks in ambulatory pediatric surgery [41]. However, only the study by Ganesh et al. [19] carried out anesthesia with ropivacaine. Finally, the age of the children was not classified in our review because these data were unclear. Therefore, the current results should be interpreted with caution.

## 5. Conclusions

We found that both TAPBs and CBs provide comparable initial pain relief and safety in children undergoing abdominal surgeries. Deciding on which block to use may depend on the clinician’s familiarity with the surgical technique and the patient’s clinical and anatomical features.

## Figures and Tables

**Figure 1 medicina-59-01527-f001:**
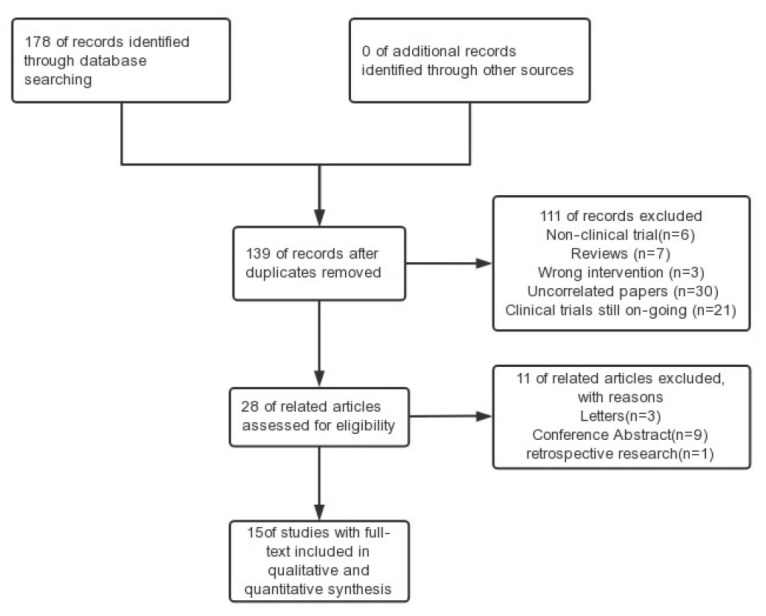
Study flow diagram.

**Figure 2 medicina-59-01527-f002:**
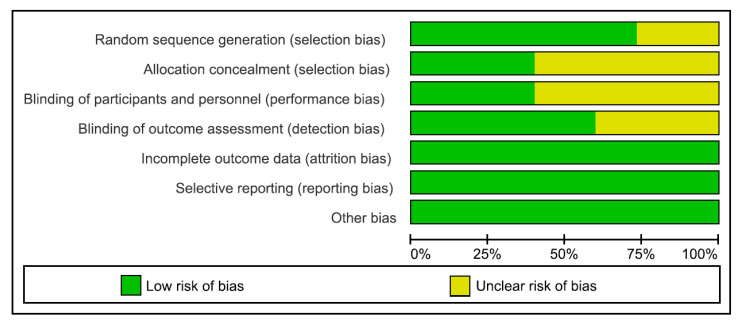
Risk of bias graph.

**Figure 3 medicina-59-01527-f003:**
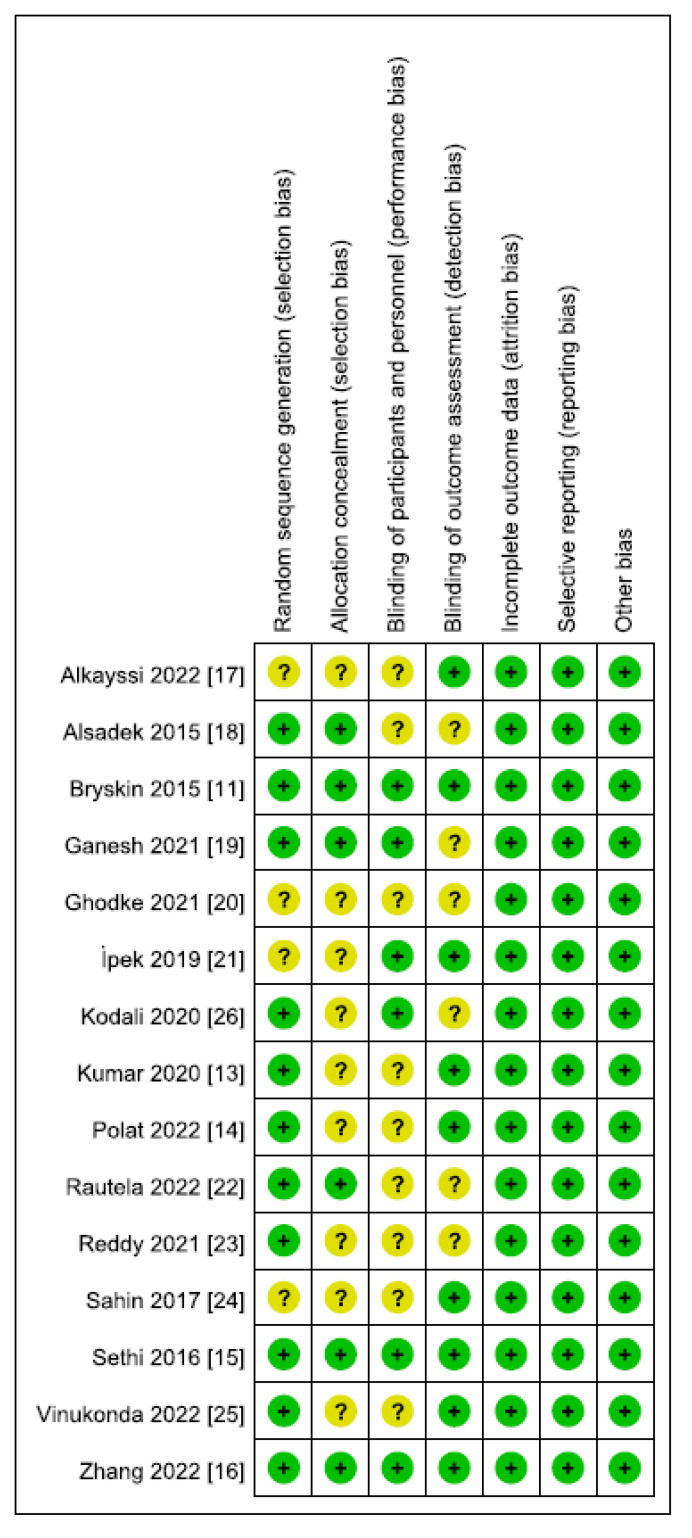
Risk of bias summary. The question mark of yellow area indicates an unclear risk of bias, and plus sign of green area indicates a low risk of bias [11,13,14,15,16,17,18,19,20,21,22,23,24,25,26].

**Figure 4 medicina-59-01527-f004:**
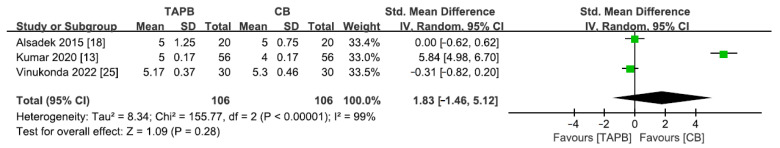
Forest plot for meta-analysis of postoperative CHEOPS score Po1h. The green square represents effect size of each study; the areas of squares are proportional to the weight given to each study. The black diamond represents the overall effect size and 95% confidence intervals [13,18,25].

**Figure 5 medicina-59-01527-f005:**
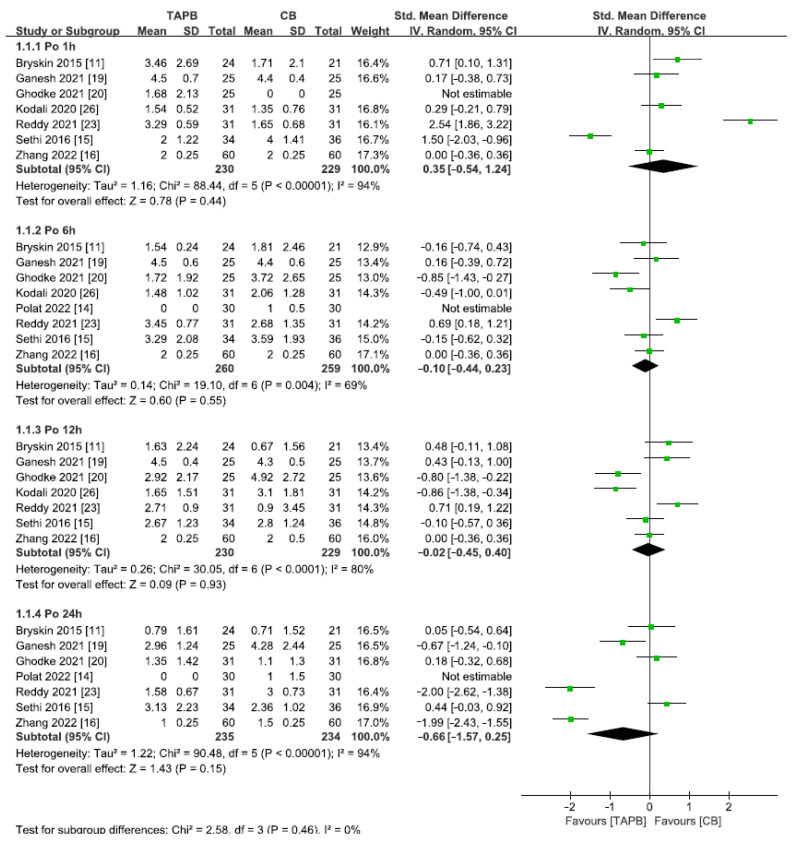
Forest plot for meta-analysis of postoperative FLACC scores. The green square represents effect size of each study; the areas of squares are proportional to the weight given to each study. The black diamond represents the overall effect size and 95% confidence intervals for each subgroup analysis [11,14,15,16,19,20,23,26].

**Figure 6 medicina-59-01527-f006:**
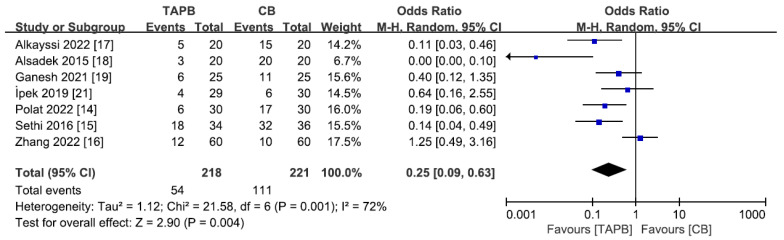
Forest plot for meta-analysis of postoperative additional analgesic requirement. The blue square represents effect size of each study; the areas of squares are proportional to the weight given to each study. The black diamond represents the overall effect size and 95% confidence intervals [14,15,16,17,18,19,21].

**Figure 7 medicina-59-01527-f007:**
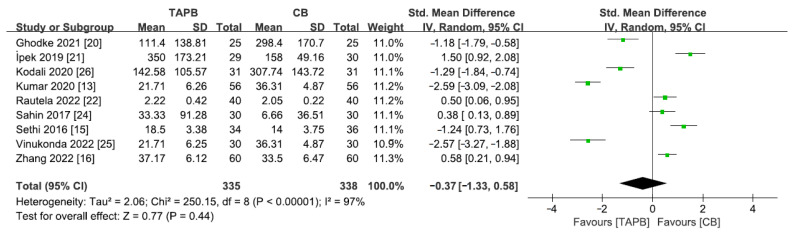
Forest plot for meta-analysis of postoperative total dose of rescue analgesic given in 24 h. The green square represents effect size of each study; the areas of squares are proportional to the weight given to each study. The black diamond represents the overall effect size and 95% confidence intervals [13,15,16,20,21,22,24,25,26].

**Figure 8 medicina-59-01527-f008:**
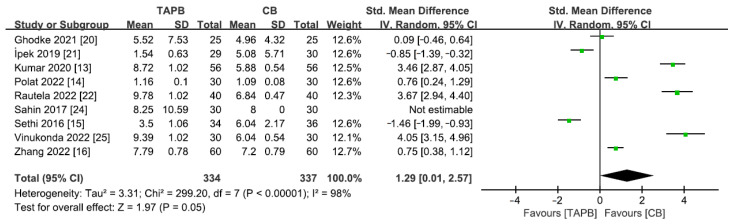
Forest plot for meta-analysis of postoperative mean duration of analgesia. The green square represents effect size of each study; the areas of squares are proportional to the weight given to each study. The black diamond represents the overall effect size and 95% confidence intervals [13,14,15,16,20,21,22,24,25].

**Figure 9 medicina-59-01527-f009:**
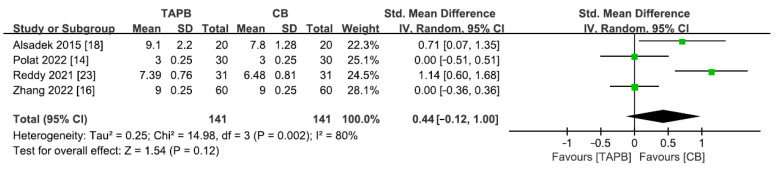
Forest plot for meta-analysis of postoperative parents’ satisfaction. The green square represents effect size of each study; the areas of squares are proportional to the weight given to each study. The black diamond represents the overall effect size and 95% confidence intervals [14,16,18,23].

**Figure 10 medicina-59-01527-f010:**
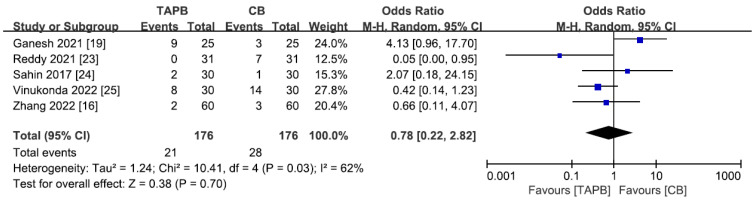
Forest plot for meta-analysis of postoperative adverse effects. The blue square represents effect size of each study; the areas of squares are proportional to the weight given to each study. The black diamond represents the overall effect size and 95% confidence intervals [16,19,23,24,25].

**Table 1 medicina-59-01527-t001:** Characteristics of studies included in the review.

Study	Location	Age (Years)	Sample Size (TAPB/CB)	Type of Surgery	Anesthetic Technology		Background Analgesia Rescue Analgesia	Follow Up	Jadad Score
					TAPB	CB			
Kumar, A. et al., 2020 [13]	India	2–8	56/56	Inguinal hernia surgery	0.5 mL/kg of 0.2% ropivacaine	1 mL/kg of 0.2% ropivacaine	Syrup paracetamol 10 mg/kg orally	24 h	6
Polat, H. et al., 2022 [14]	Turkey	1–7	30/30	Pediatric inguinal hernia repair surgeries	0.5 mL/kg of 0.25% bupivacaine	1 mL/kg bupivacaine 0.25%	Paracetamol IV or per oral was administered	24 h	7
Sethi, N. et al., 2016 [15]	India	2–6	34/36	Lower abdominal surgery	0.5 mL/kg of 0.25% bupivacaine	Bupivacaine (0.75 mL/kg of 0.25%)	Oral paracetamol 20 mg/kg		4
Zhang, Y. et al., 2022 [16]	China	1–12	60/60	Lower abdominal laparoscopic surgery	Ropivacaine	Ropivacaine	Tramadol	24 h	7
Alkayssi, H. et al., 2022 [17]	Iraq	3–9	20/20	Post-inguinal surgery (elective inguinal hernia repair surgery)	Bupivacaine 0.25% (0.5 mL/kg)	Bupivacaine 0.25% (1 mL/kg)	Paracetamol 10 mg/kg	12 h	5
Alsadek, W.M. et al., 2015 [18]	Egypt	2–7	20/20	Lower abdominal surgeries	A bolus of 0.5 mL/kg bupivacaine 0.25%	1.0 mL/kg bupivacaine 0.25%	Paracetamol (acetaminophen) suppository 15 mg/kg	12 h	6
Bryskin, R.B. et al., 2015 [11]	America	1–9	24/21	Lower abdominal surgery	0.5 mL/kg of 0.25% bupivacaine + epinephrine 1:200,000	1 mL/kg of 0.25% bupivacaine + epinephrine 1:200,000	Morphine 0.05 mg/kg IV every 2 h as needed for moderate to severe pain	24 h	7
Ganesh, B. et al., 2021 [19]	India	2–7	25/25	Infraumbilical surgeries	0.5 mL·kg^−1^ of 0.2% Ropivacaine	1 mL·kg^−1^ of 0.2% ropivacaine	Injection paracetamol 20 mg/kg, intravenous fentanyl 1 μg/kg	24 h	7
Ghodke, S.M. et al., 2021 [20]	India	1–8	25/25	Extraperitoneal lower abdominal surgeries	0.5 mL/kg of 0.2% levobupivacaine and 0.1 mg/kg dexamethasone.	1 mL/kg of 0.2% levobupivacaine and 0.1 mg/kg dexamethasone	Paracetamol 15 mg/kg IV	24 h	4
İpek, C.B. et al., 2019 [21]	Turkey	0.5–14	29/30	Lower abdominal surgery	0.5 mL kg^−1^ of 0.25% bupivacaine solution	0.5 mL kg^−1^ of 0.25% bupivacaine solution	10 mg kg^−1^ of ibuprofen syrup	24 h	5
Rautela, M.S. et al., 2022 [22]	India	3–10	40/40	Unilateral infraumbilical surgery	0.5 mL/kg of 0.25% bupivacaine	0.75 mL/kg of 0.25% bupivacaine	Syrup paracetamol 15 mg/kg body weight or injection paracetamol 10 mg/kg body weight	24 h	6
Reddy, A. et al., 2021 [23]	India	2–10	31/31	Lower abdominal surgeries	0.25% bupivacaine 0.5 mL/kg with 1 µg/kg dexmedetomidine.	0.25% bupivacaine 0.5 mL/kg with 1 µg/kg dexmedetomidine	NM	24 h	6
Sahin, L. et al., 2017 [24]	America	1–7	30/30	Unilateral lower abdominal surgery	0.25% levobupivacaine added to 1/200,000 adrenaline at a dose of 0.5 mL/kg	0.25% levobupivacaine added to 1/200,000 adrenaline at a dose of 0.7 mL/kg	NM	24 h	4
Vinukonda, M.K. et al., 2022 [25]	India	2–8	30/30	Unilateral open inguinal hernia repair	0.5 mL/kg 0.25% bupivacaine	0.75/kg 0.2% bupivacaine	Paracetamol IV 15 mg/kg	24 h	4
Kodali, V.R.K. et al., 2020 [26]	India	0.5–8	31/31	Inguinal hernia repair	0.5 mL·kg^−1^ of 0.25% bupivacaine	1 mL·kg^−1^ of 0.25% bupivacaine	Intravenous paracetamol was given at a dose of 7.5 mg·kg^−1^ for children weighing < 10 kg and 15 mg·kg^−1^ for children weighing > 10 kg	24 h	6

TAPB, transversus abdominis plane block; CB, caudal block; NM, not mentioned.

**Table 2 medicina-59-01527-t002:** Summarizes effect estimates for all of the outcomes included in this meta-analysis.

Outcome	Studies	Participants	Statistical Method	Effect Estimate (95% CI)	*p*-Value
Pain score at 1 h (CHEOPS score)	3	212	SMD (IV,RE)	1.83 (−1.46 to 5.12)	0.28
Pain score at 1 h (FLACC score)	7	459	SMD (IV,RE)	0.35 (−0.54 to 1.24)	0.44
Pain score at 6 h (FLACC score)	8	519	SMD (IV,RE)	−0.10 (−0.44 to −0.23)	0.55
Pain score at 12 h (FLACC score)	7	459	SMD (IV,RE)	−0.02 (−0.45 to −0.40)	0.93
Pain score at 24 h (FLACC score)	7	469	SMD (IV,RE)	−0.66 (−1.57 to −0.25)	0.15
Additional analgesic requirement	7	439	OR (MH,RE)	0.25 (0.09 to 0.63)	0.004
Total dose of rescue analgesic given in 24 h	9	673	SMD (IV,RE)	−0.37 (−1.33 to −0.58)	0.44
The mean duration of analgesia	9	671	SMD (IV,RE)	1.29 (0.01 to 2.57)	0.05
Parents’ satisfaction	4	282	SMD (IV,RE)	0.44 (−0.12 to 1.0)	0.12
Adverse effects	5	352	OR (MH,RE)	0.78 (0.22 to 2.82)	0.70

Effect estimates. SMD, standardized mean difference; IV, inverse variance; RE, random effects; MD, mean difference; OR, odds ratio; MH, Mantel–Haenszel.

## Data Availability

Not applicable.

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
