# Peer review of "A Systematic Review and Meta-Analysis Comparing the Effectiveness of Transversus Abdominis Plane Block and Caudal Block for Relief of Postoperative Pain in Children Who Underwent Lower Abdominal Surgeries"

_medicina, 2023, doi:10.3390/medicina59091527_

Round 1

Reviewer 1 Report

1. This paper is very important from the point of view of the practical use of two different types of analgesia.
2. The authors compare two completely different perioperative analgesia techniques in terms of efficacy and level of performance, which they themselves state. The authors do not justify this comparison.

3. the abstract of the paper is inadequately written; it mentions TAPB and CB as two peripheral analgesia techniques and at the same time states that they are inadequate for visceral pain. They do not mention the importance of other perioperative analgesia techniques, which should be added to the two described techniques for an accurate evaluation or comparison. 

4. The authors do not report the complication rates for both types of anaesthesia, nor the selection of patients for each type of block. The quality of the presentation is lost.

5. Both types of block are used in clinical practice, as confirmed by the studies analysed by the authors. The authors somewhat awkwardly add that they want to demonstrate the justification for the use of TAPB and CB, so this part of the paper needs to be redrafted and corrected.

6. The studies analysed by the authors were mostly related to lower abdominal wall or inguinal hernia surgery; the comparison with major mid- and upper abdominal wall surgery is a different issue. The question arises as to the meaningfulness of the comparison. Moreover, the use of the different analgesics cited by the authors of the studies reviewed is completely different. There is a question of the objectification of the results.

English shoud be improved.

Reviewer 2 Report

I want to be sure that I understood well: both LRA were administered after surgery? If yes, it means that in all the paper that you reviewed the method was the same and the only differences was about the choice of the LA?

The age of pediatric pts, in your opinion, can influence the outcome of the LRA ( older children may have already experienced pain!)

The pediatric anesthesia is a particular situation that can interest a specialize group of anesthetists, is not easy to  conduct a research and for this is very difficult to compare papers that cannot use the same methods. So all the results that can be argumented are well received, because the difficulty is known.

Sometimes in an adult pt we perform LRA before surgery because the clinical risk is high and we need to reduce the pharmacological risk using less drugs, can be the same for pediatric pts?

In the opinion of the authors its a cultural problem to choose one or the other LRA? Because the rule everywhere is: use what you know better! But with this approach never can change, so we need to open our minds and change our behaviors?

The purpose of their paper is to analyze which is the best and safer anesthetic approach in paediatric pts undergoing abdominal surgery?

Reviewer 3 Report

The manuscript is welk written and presents data in an adequate way.

The discussion is brief and focuses on Main issues.

As alwaays for mera analysis I recommend to habe an expert in statistics as additional (in House referee) since my statistical experienceas clinician is not enough for such a highly sophisticated Evaluation.

if statistic is Sound conclusions are in according with results.

in figures of the results section it is stated on the X axis „ favours results vs favours experimental“. I think that is automatically generated by the programm. As reader I would prefer : „favours  ACB vs favours TAPB“

apart from this very Minor Revision I suggest to accept the manuscript in its present Form.

Round 2

Reviewer 1 Report

1. Authors need to better describe the type of surgical procedures and laparotomies that are associated with pain, otherwise, the circumstances remain underestimated and incomprehensible.

2. Authors should address incisional wounds and laparotomies and TAPB and CB.

3. There is a large variability in the delivery or identification of pain; an analysis by age would be needed to obtain significantly more relevant information on both the choice of technique and the need for painkillers.

4. Can the authors select similar studies to reduce bias, as they report different amounts of analgesics and the route of TAPB and CB. What would such a change mean for reporting and correct outcome assessment?

5. What is the rational basis for the choice of the observation time between extubation and the first call for analgesia? The authors do not provide a precise postoperative protocol or a standard postoperative analgesic regimen.

6. In the discussion on patient satisfaction, the authors should define more precisely the age of the subjects in the studies or divide the patients into age groups.  

Editing is mandatory.
